# Optimization of Monobenzone-Induced Vitiligo Mouse Model by the Addition of Chronic Stress

**DOI:** 10.3390/ijms24086990

**Published:** 2023-04-10

**Authors:** Jing Dong, Yifan Lai, Xiaofeng Zhang, Yunyun Yue, Hui Zhong, Jing Shang

**Affiliations:** 1School of Traditional Chinese Pharmacy, China Pharmaceutical University, Nanjing 211198, China; dongjingtx@gmail.com (J.D.);; 2State Key Laboratory of Natural Medicines, China Pharmaceutical University, Nanjing 210009, China; 3Jiangsu Key Laboratory of TCM Evaluation and Translational Research, China Pharmaceutical University, Nanjing 211198, China; 4NMPA Key Laboratory for Research and Evaluation of Cosmetics, National Institutes for Food and Drug Control, Beijing 100050, China

**Keywords:** vitiligo, chronic unpredictable mild stress, monobenzone, metabolomics

## Abstract

Vitiligo is a common primary, limited or generalized skin depigmentation disorder. Its pathogenesis is complex, multifactorial and unclear. For this reason, few animal models can simulate the onset of vitiligo, and studies of drug interventions are limited. Studies have found that there may be a pathophysiological connection between mental factors and the development of vitiligo. At present, the construction methods of the vitiligo model mainly include chemical induction and autoimmune induction against melanocytes. Mental factors are not taken into account in existing models. Therefore, in this study, mental inducement was added to the monobenzone (MBEH)-induced vitiligo model. We determined that chronic unpredictable mild stress (CUMS) inhibited the melanogenesis of skin. MBEH inhibited melanin production without affecting the behavioral state of mice, but mice in the MBEH combined with CUMS (MC) group were depressed and demonstrated increased depigmentation of the skin. Further analysis of metabolic differences showed that all three models altered the metabolic profile of the skin. In summary, we successfully constructed a vitiligo mouse model induced by MBEH combined with CUMS, which may be better used in the evaluation and study of vitiligo drugs.

## 1. Introduction

Vitiligo is a common primary, limited or generalized skin depigmentation disease. It is caused by the reduction or loss of functional melanocytes in the skin and hair follicles, and it affects nearly 100 million people worldwide [1,2]. As this skin disorder manifests itself in altered and impaired appearance, the color differences manifested in exposed and sensitive areas often lead to high psychological stress, decreased quality of life and a strong desire for treatment [3]. The pathogenesis mainly includes genetic, autoimmune, mental and oxidative stress, endocrine dysfunction, and melanocyte shedding [4,5,6,7], but its exact pathogenesis is not yet clear, and its development is somewhat individualized and unpredictable. Vitiligo is one of the four major dermatological diseases that are easy to diagnose and difficult to treat [8,9]. In recent years, psychosomatic factors have been found to play an important role in the development of skin diseases, and mental stress can induce or worsen certain skin diseases [10]. Modern medicine has confirmed that many skin diseases such as vitiligo, urticaria, eczema, neurodermatitis, psoriasis, and baldness are closely related to psychosomatic factors in their etiology, pathogenesis, and disease evolution, and these skin diseases are also referred to as psychiatric-associated skin diseases [11,12,13]. In 2018, Vallerand et al. evaluated the Health Improvement Network database and found a 64% increased risk of vitiligo (HR = 1.64, 95% CI: 1.43–1.87, *p* < 0.0001) [14,15,16,17,18] in patients with a major depressive disorder (n = 405,397) compared to a control group (n = 573,9048). Their findings were highly suggestive of a possible pathophysiological link between depression and the development of vitiligo. In contrast, the current international studies on vitiligo have focused more on the effects of autoimmune, inflammatory, and oxidative stress factors of the disease and have hardly taken psychiatric factors into account.

Nowadays, the animal models used in the drug development of vitiligo treatment can be broadly classified into three categories: spontaneous vitiligo models, chemically induced vitiligo models, and transgenic vitiligo animal models. Among them, spontaneous vitiligo models are often unstable and are not easily available. Transgenic animal models include TCR transgenic mouse models of PMEL, TRP, or TYR, and these mouse models of vitiligo induced by immunity are mainly used for studies related to immune mechanisms [8]. Chemically induced vitiligo animal models mainly include hydrogen peroxide-induced, hydroquinone-induced, and monobenzone (MBEH)-induced models of depigmentation [19,20,21]. MBEH is a topical depigmentation agent that breaks down melanin and prevents melanin production in the skin, and its ubiquitination of tyrosinase and autophagy of melanosomes can directly induce CD8^+^ T cell activation. These T cells attack melanocytes leading to depigmentation [22,23,24]. MBEH is a derivative of hydroquinone, which is less toxic, less irritating to the skin, mild in action, and a better choice for chemical depigmentation [25]. MBEH is converted to 4-benzo-1,2-benzoquinone in vivo and combines with cysteine residues of tyrosine to form quinone semi-antigens [26], which is consistent with the autoimmune hypothesis of vitiligo. Not only does MBEH induce oxidative stress in pigment cells and induce lysosomal degradation of melanin through autophagy, but it also secretes inducible HSP70 during cellular stress [27], similar to the mechanism of oxidative stress in vitiligo, which leads to lipid peroxidation in melanocytes and protein structure-function variation to enhance the autoimmune response.

Because the pathogenesis of vitiligo is unknown and may be caused by multiple factors, there is not yet an animal model that fully represents the clinical features of vitiligo. Moreover, no animal model has been reported in which psychiatric factors are associated with vitiligo disease. Chronic unpredictable mild stimulation (CUMS) and chronic restraint stress are commonly used methods for modeling chronic mental stress, but CUMS is more consistent with the chronic stress state experienced by humans daily. In a previous study, we found that mental stress induces the release of neuropeptides [28,29,30], affects the central hypothalamic-pituitary-adrenal axis [31], and results in depigmentation or hair growth inhibition. Therefore, we selected the MBEH and CUMS models to construct a multifactor-induced depigmentation mouse model. The changes in skin melanin were preliminarily analyzed in the pathological section. Metabolic differences were then compared by metabolomics. Finally, we successfully constructed a stress-induced vitiligo mouse model, which can be used for the further study of stressed vitiligo.

## 2. Results

### 2.1. Chronic Stress Suppresses the Melanogenesis in Skin

According to our previous studies, mental stress results in depigmentation or hair growth inhibition [28,29,30]. Therefore, we firstly observed changes in melanin in the CUMS model. The flow chart of CUMS modeling is shown in Figure 1A. Body weight is one of the most intuitive indicators that reflects the stress state of mice. As shown in Figure 1B, the body weight of mice in the control group increased steadily with the extension of time. The body weight was lower in the CUMS group (*p* < 0.001), but there was also a trend of fluctuating and a very slow increase over time. The results suggest that chronic stress could significantly inhibit weight gain. The thymus and the spleen are two important immune organs involved in immune function. The spleen is involved in cellular and humoral immunity. The thymus is mainly involved in the development of T-lymphocytes and the secretion of hormones such as thymosin and thymopoietin, which play an important role in cellular immune function. Generally, when the body is subjected to adverse stimuli, the thymus index and spleen index will change accordingly, thus affecting the immune function of the body [32]. As shown in Figure 1C, chronic stress had no significant effect on the thymus index (*p* > 0.05), whereas the spleen index increased significantly (*p* < 0.05), suggesting that chronic stress may lead to the activation of the immune system. Behavioral indicators are the key indicators that reflect the state of stress [33]. Chronic stress significantly reduced the total moving distance (number of frames climbed, *p* < 0.001), grooming frequency (*p* < 0.001), and rearing frequency (*p* < 0.05) (Figure 1D), suggesting a significant reduction in spontaneous activity. Chronic stress significantly increased the total immobility time in the forced swimming test (FST) and tail suspension test (TST) (*p* < 0.05) (Figure 1E), suggesting that chronic stress can lead to behavioral despair and depression-like behavior in mice.

Further, the effects of chronic mental stress on skin melanin synthesis in mice were investigated. On the 28th day of chronic stress, the back skin was depilated to induce the hair follicles to enter the anagen phase, during which melanocytes were formed and melanin synthesis was initiated. There was no significant difference in pigmentation on day one post-depilation (PD1). On PD8 and PD12, it was found that the control group had darker coloration in the dorsal skin, whereas the dorsal coloration of stressed mice was significantly diminished (*p* < 0.01, *p* < 0.001), suggesting that chronic stress significantly inhibited pigment synthesis (Figure 1F). The result of H&E staining showed that the melanin granules in the growing hair follicles of mice were significantly reduced after CUMS (Figure 1G). Tyrosinase (TYR) is the rate-limiting enzyme involved in the process of melanin synthesis [34]. Microphthalmia-associated transcription factor (MITF), a transcription factor of TYR, can promote the expression of TYR [35,36,37]. As shown in Figure 1H, the expression levels of MITF and TYR proteins were significantly down regulated after CUMS compared with CON (*p* < 0.01, *p* < 0.01). In summary, these results suggest that chronic mental stress inhibited the ability of melanin synthesis.

### 2.2. MBEH Combined with CUMS (MC) Leads to Depression-like Behavior in Mice

According to the above results, stress is also an inducement of depigmentation. Thus, chronic mental stress was added to induce depigmentation together with MBEH (Figure 2A). The body weight of mice in the MBEH group was consistent with that in the control group (no statistical difference was found), suggesting that MBEH alone did not affect the body weight of mice. In addition to MBEH, additional chronic stress began to reduce body weight in the second round of modeling and at the end of modeling (*p* < 0.001), which was consistent with the results of chronic stress alone (Figure 2B). The thymic index was significantly lower (*p* < 0.01) in the MBEH group compared with the control group, whereas the spleen index was not significantly different but tended to increase. There were no significant changes in thymus and spleen index in the mice of the MC group (Figure 2C). Behaviorally (Figure 2D,E), there were no significant changes in the total moving distance (number of frames climbed), grooming frequency, rearing frequency, and total TST immobility time in the MBEH group, whereas the total FST immobility time was significantly reduced (*p* < 0.001), suggesting that MBEH did not affect the behavior of the mice. Compared with the MBEH group, the spontaneous activity and total FST immobility time of mice in the MC group were significantly decreased (*p* < 0.05), suggesting that mice exhibit depression-like behavior as a result of chronic stress.

### 2.3. Mice in the MC Group Showed More Inhibition in Melanogenesis

Next, the effects of MBEH and MC on skin melanin synthesis in mice were investigated. As shown in Figure 3A, the number of white hairs on the backs of the mice could not be accurately determined by visual observation, so a multi-probe skin testing system was used to record the black and white hairs at a deep level and the photos were scanned in grayscale using ImageJ software. The results showed that, compared with CON, MBEH alone and MC could significantly induce depigmentation (*p* < 0.01, *p* < 0.001). Compared to the MBEH group, the addition of chronic stress significantly aggravated depigmentation in mice (*p* < 0.05) (Figure 3B). Melanin content in the hair follicles was significantly reduced in the MBEH and MC groups, and a small amount of follicular melanin was completely absent (Figure 3C). Compared with the MBEH group, the melanin deficiency was more severe after the addition of chronic mental stress. Furthermore, compared with CON, MBEH and MC significantly decreased the expression levels of MITF and TYR. The addition of chronic mental stress further significantly inhibited the expression levels of MITF and TYR proteins (*p* < 0.05) (Figure 3D).

MBEH is metabolized by pigment cells to form semi-antigens, which promote the production of melanocyte-specific T lymphocytes, thereby inducing an autoimmune response. CD8^+^ T cells are cytotoxic T lymphocytes, which are responsible for the autoimmune destruction of melanocytes in vitiligo. Therefore, by examining the infiltration of CD8^+^ T cells in the skin of mice, the reliability of the model was investigated from the perspective of immunity. Compared with CON, the number of CD8^+^ T cells in the skin was increased in all three models. Although there was no infiltration of CD8^+^ T cells in the CUMS group, aggregates were present around the vasculature of the skin (Figure 3E). In the MBEH group, T cells were mainly distributed in the epidermis. By giving additional chronic stress, the number of CD8^+^ T cells was significantly increased compared to the MBEH group, and there was a significant accumulation in both the dermis and epidermis (Figure 3F), indicating that chronic mental stress can induce and exacerbate the immune response. In conclusion, these results suggest that MC exacerbates the degree of depigmentation.

### 2.4. Metabolic Analysis

The alteration of the metabolic profile is the most direct manifestation of disease [38]. The metabolic profiles of skin in CUMS, MBEH and MC mice were analyzed. The sPLSDA can effectively reduce the number of variables or metabolites in the high-dimensional metabolome, resulting in stable and easily understood models. The sPLSDA analysis method was used to analyze the metabolite profile of the skin. All three models were well separated from the CON, and all significantly altered the metabolic profiles (Figure 4).

Using FC (Fold change) > 1.5 and FDR < 0.05 as criteria, volcano and heat maps of the differential metabolites in the skin of the three models were drawn, and 184, 183, and 237 differential metabolites were screened out, respectively (Figure 5).

KEGG enrichment analysis was performed on the differential metabolites. As shown in Figure 6, compared with CON, CUMS significantly altered the energy metabolism-related pathways, which could regulate the levels of various inflammatory mediators such as IL-1β, IL-6 and IL-17, etc. MBEH significantly altered the metabolic pathways involved in inflammation and immune response and cellular oxidative stress. MC significantly altered the metabolic pathways regulating cellular oxidative stress and pigment synthesis.

## 3. Discussion

Nowadays, stress shows a significant impact on people’s health. The skin, as a stress organ, communicates in a complex way with the brain through neural and humoral signals [39]. The occurrence and development of many skin disorders are closely related to psychological factors. Clinically, the incidence of mental diseases among dermatological patients is estimated to be from 30% to 60% [40]. Modern medicine has demonstrated that many dermatological diseases such as vitiligo [10], alopecia areata [12,41], pruritus, urticaria, psoriasis [42,43], neurodermatitis [44], psoriasis [45] are strongly associated with psychological factors. Clinical studies have found that patients with vitiligo have a significantly increased risk of multiple mental diseases compared with healthy individuals. The total prevalence of anxiety in patients with vitiligo is as high as 35.8%, and there is a statistically significant difference in the anxiety rate between female and male patients (47.32% vs. 42.4%) (*p* < 0.05) [46]. It is suggested that mental stress is always accompanied by the evolution of vitiligo [39]. Therefore, we investigated the effect of chronic stress on skin pigment in mice by administering CUMS (Figure 1A). Body weight monitoring and behavioral detection were used to evaluate the success of the stress model. The results of the open-field test (OFT), forced swimming test (FST), and tail suspension test (TST) showed that the spontaneous activity of mice in the CUMS group was weakened, and they were in a state of behavioral despair (Figure 1B–E). Phenotypically, a significant depigmentation could be observed in PD8 and PD12 (Figure 1F). Pathologically, the number of melanin particles in the hair follicle was significantly decreased after CUMS (Figure 1G). Biochemically, CUMS inhibited the expression of MITF and TYR, melanin synthesis-related proteins (Figure 1H). These results suggest that chronic stress can inhibit melanin synthesis.

MBEH can be metabolized by melanocytes, thus making melanocyte-specific proteins such as tyrosinase hemi-antigenic, which in turn leads to cytotoxic autoimmunity against pigment cells and induces vitiligo. After entering melanocytes, MBEH is metabolized by tyrosinase to form superoxide ions, which then form hydrogen peroxide to promote the destruction of lipid and protein structures to form specific antigens, and further induce vitiligo [47]. By comparing the existing vitiligo models, it was found that MBEH-induced depigmentation models were most similar in terms of phenotype and pathological mechanism. Due to the influence of the hair follicle cycle, the model of MBEH reported in the literature was modified, and 40% MBEH was applied [48]. On this basis, CUMS was also administered to construct the vitiligo model induced by chronic mental stress (Figure 2A). MBEH did not affect the behavior of mice. Mice exhibited depression-like behavior after the addition of CUMS (Figure 2B–E). The whiteness of the hair on the backs of the MBEH and MC mice was significantly increased under a multi-probe skin testing system. Compared with MBEH alone, the depigmentation of the mice in the MC group was more significant (Figure 3A,B). The pathological results showed that the lack of pigment particles in the mouse hair follicles was aggravated, suggesting that chronic mental stress could exacerbate the symptoms of chemically induced depigmentation (Figure 3C). The expression of MITF and TYR was inhibited in all three models, and the reduction of MITF and TYR expression levels was aggravated by the addition of chronic stress (Figure 3D). All three models can promote the infiltration of CD8^+^ T cells in the skin, and the number of T cells increased significantly and the infiltration range increased after the addition of chronic mental stress (Figure 3E,F). The above results reflect the role of chronic mental stress in promoting depigmentation, suggesting that CUMS can aggravate the chemically induced depigmentation symptoms.

Finally, the effects of modeling on the metabolic profile of mouse skin were investigated. The results showed that there were significant differences in the metabolic profiles of the three models of mice compared to the control group (Figure 4 and Figure 5). CUMS significantly altered the energy metabolism-related pathways. MBEH significantly altered the metabolic pathways involved in inflammation and immune response and cellular oxidative stress. MC significantly altered the metabolic pathways regulating cellular oxidative stress and pigment synthesis (Figure 6). The above results reflect the role of chronic mental stress in promoting depigmentation, and the establishment of a multifactor-induced animal model combined with chronic mental stress has more guiding significance for the evaluation of vitiligo drugs. In summary, we optimized the MBEH model and successfully constructed a mouse model of chronic mental-stress-induced vitiligo.

## 4. Materials and Methods

### 4.1. Mice

Five-week-old male C57BL/6 mice weighing 18 ± 22 g were purchased from Changzhou Cavens Laboratory Animal Co., Ltd. The animals were housed in the Pharmaceutical Animal Experimental Center, China Pharmaceutical University, in a temperature-controlled room (20–25 °C) with a standard 12 h light and 12 h dark cycle. The mice were provided with unrestricted food and water unless otherwise stated.

### 4.2. CUMS Model

In the CUMS group, mice were given eight different stimuli for 40 days. The stimulus included: food deprivation (12 h), water deprivation (12 h), wet litter (250 mL of water on the litter bed, 12 h), cage shaking (15 min), cage tilt (30°, 24 h), cold water swimming (5 min), clipped tail (2 cm from the tip of the tail, 2 min), and restraint (2 h). The mice were modeled by alternating one or two methods each day, and each method was used discontinuously for three days so that the mice could not predict and adapt to the stimulus.

### 4.3. MBEH Model

In the MBEH group, the hair follicles were induced to enter the anagen phase by rosin paraffin waxing. On the 5th day after hair removal, a continuous application of 40% MBEH (50 mg per day, CAS: 103-16-2, Sigma, St. Louis, MO, USA) was started and stopped after two weeks. After the hair follicles had re-entered the resting phase, rosin paraffin waxing was used to induce the hair follicles to enter the anagen phase, and the procedure was repeated for three rounds. The flowchart of the MBEH model is shown in Figure 2A.

### 4.4. MBEH Combined with CUMS (MC) Model

Mice in the MC group were treated in the same way as mice in the MBEH group. However, CUMS was given at the same time as the application of 40% MBEH. The flowchart of the MC model is shown in Figure 2A.

### 4.5. Behavior Test

The open-field test (OFT), forced swimming test (FST), and tail suspension test (TST) were carried out as previously described [31,49]. The OFT was used to observe the behavioral changes in each group of mice and to test the spontaneous activity of the mice. The mice were placed in the center of an open-field experimental chamber (40 × 40 × 30 cm). The FST and TST were used to measure the degree of desperation. During the FST, mice were placed alone in a clear plastic beaker with a diameter of 10 cm and a height of 30 cm. The beaker contained water of 24 ± 2 °C with a depth of 20 cm. In the TST, the mice were taped 1 cm away from the tail tip with the head facing down about 20 cm from the bottom. After 2 min of adaptation, the movement trajectory and immobility time of the mice were analyzed for 4 min.

### 4.6. Hematoxylin-Eosin Staining

The skins of mice were embedded in paraffin. The embedded tissue was cut into 4 μm-thick slices and adhered to the slides and dried at 45 °C for 4 h. The sections were stained with hematoxylin and eosin and photographed [29].

### 4.7. Immunofluorescence

Immunofluorescence analysis was conducted as described previously [29,31]. The slides of skin were stained with primary antibodies against CD8 (Abcam, Cambridge, UK, ab217344) and left overnight at 4 °C. After incubation with the secondary antibody, cell nuclei were counterstained with DAPI (Solarbio, Beijing, China). Immunosignals were detected and analyzed using a Carl Zeiss LSM700 confocal microscope (Oberkochen, German).

### 4.8. Western Blotting (WB) Analysis

An amount of 30 mg of skin was cut and ground into powder at a low temperature and placed in 100 μL IP lysate containing protease inhibitor PMSF (Beyotime Biotechnology, Shanghai, China). The supernatant was collected and centrifuged at 12,000 rpm for 5 min at 4 °C. A total of 30 μg of proteins was isolated with 10% SDS-PAGE gel and transferred to a nitrocellulose membrane, before being sealed with 5% skimmed milk powder at room temperature for 2 h and then washed with TBS (0.1% Tween-20 in PBS) three times. The proteins were combined with antibodies against MITF (Santa Cruz, CA, USA, sc-56725), TYR (Santa Cruz, CA, USA, sc-15341), and GAPDH (CST, Beverly, MA, USA, 5174) and left at 4 °C overnight. After incubation with a secondary antibody, the proteins were visualized using an enhanced chemiluminescence detection system (Tanon, Shanghai, China). The WB results represent at least three independent experiments.

### 4.9. Liquid Chromatography-Mass Spectrometer (LC-MS) Analysis

Chromatographic separation was performed on a high-performance liquid chromatography-quadrupole time-of-flight mass spectrometer (Agilent 1260, Agilent, Santa Clara, CA, USA). The determination was performed on a Waters XSelect HSS T3 column (2.1 × 100 mm, 2.5 μm) at a temperature of 40 °C. The mobile phase consisted of 0.1% formic acid (A) (Fisher, Waltham, MA, USA) and acetonitrile (B), and the gradient was run at a flow rate of 0.5 mL/min, eluting in 100% A for 4 min, then increasing solvent B to 60% over 16 min, and eluting in 100% solvent B for 8 min. The injection volume was 5 μL per sample. MS data were obtained from a mass spectrometer equipped with an electrospray source in negative mode, and the sweep range was 50–1000 m/z. The mass spectrometry parameters were set as follows: gas temperature of 325 °C, gas flow rate of 8 L/min, nebulizer pressure of 40, sheath gas temperature of 350 °C, sheath gas flow rate of 12 L/min, 500 nozzle voltage for negative, debris of 130 volts, skimmer of 65, and octopole RF peak of 750.

The LC-MS data files were converted to mzXML format files for further analysis using the Proteowizard software (version: 3.0.19291-5e92459cc). Retention time correction and ion peak matrix extraction were performed using the xcms R package (version: 3.10.2). After obtaining the ion matrix, univariate statistical analyses were performed, including fold change (FC), *t*-test, and volcano plot analysis. Multivariate statistical analyses were performed using R language, and those ions with FC > 1.5 and *p* < 0.05 were screened out. The metabolite profiles of the skin tissues were analyzed by using Sparse PLS discriminant analysis (sPLSDA). KEGG pathway enrichment analysis of differentially metabolized ions was performed using the MetaboAnalystsR package (version 3.0) to investigate the functional distribution of the model.

### 4.10. Statistical Analysis

All data are presented as the mean ± standard deviation from at least three independent experiments. All statistical analyses were performed using a paired two-tailed Student’s t-test or one-way analysis of variance for multiple groups using Graphpad Prism7.0. A *p*-value of <0.05 denoted statistical significance.

## Figures and Tables

**Figure 1 ijms-24-06990-f001:**
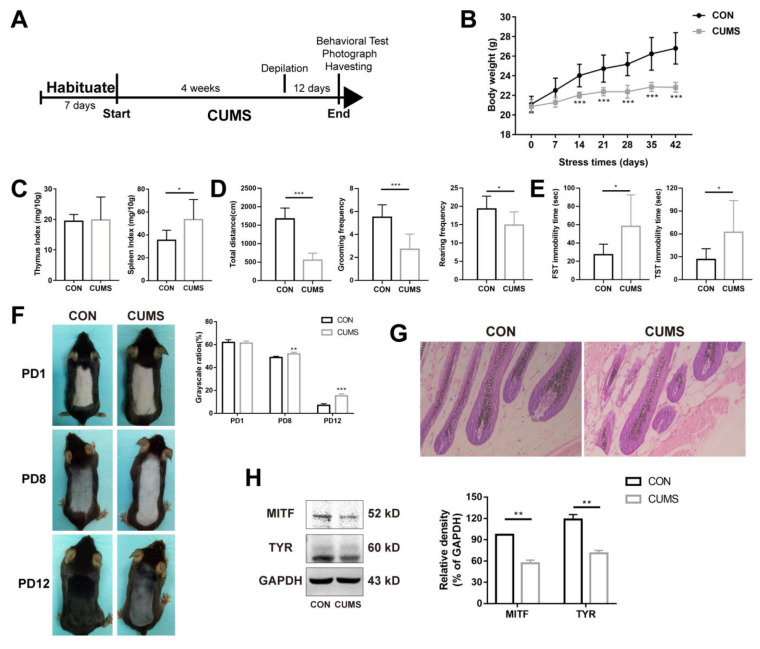
Chronic stress suppresses melanogenesis in the skin (n = 9). (**A**) The flowchart of CUMS modeling. (**B**) The effect of chronic stress on body weight. (**C**) The effect of chronic stress on the thymus index and spleen index. (**D**,**E**) The effect of chronic stress on spontaneous activity and behavioral despair in mice. (**F**) Effects of chronic stress on melanin synthesis in the back skin of mice. The grayscale ratio = mean grayscale/255. (**G**) H&E staining of the mouse skin. Magnification: 200×. (**H**) The protein level of MITF and TYR were determined by Western blot analysis. Results were shown as mean ± standard deviation (SD) and were representative of three independent experiments. * *p* < 0.05, ** *p* < 0.01, *** *p* < 0.001, compared with CON. Two groups were tested using a *t*-test.

**Figure 2 ijms-24-06990-f002:**
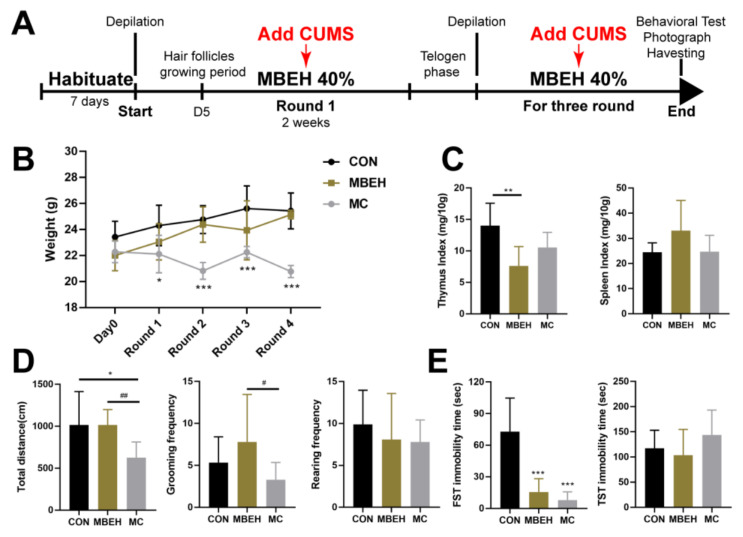
Mice in the MC group exhibit depression-like behavior (n = 9). (**A**) The flowchart of MBEH and MC modeling. (**B**) Effects of MBEH and MC on body weight. (**C**) The effects of MBEH and MC on the thymus index and spleen index. (**D**,**E**) The effect of MBEH and MC on spontaneous activity and behavioral despair in mice. Data were presented as the mean ± SD. * *p* < 0.05, ** *p* < 0.01, *** *p* < 0.001, compared with CON; ^#^
*p* < 0.05, ^##^
*p* < 0.01, MC was compared with MBEH. Multiple groups were tested using one-way ANOVA.

**Figure 3 ijms-24-06990-f003:**
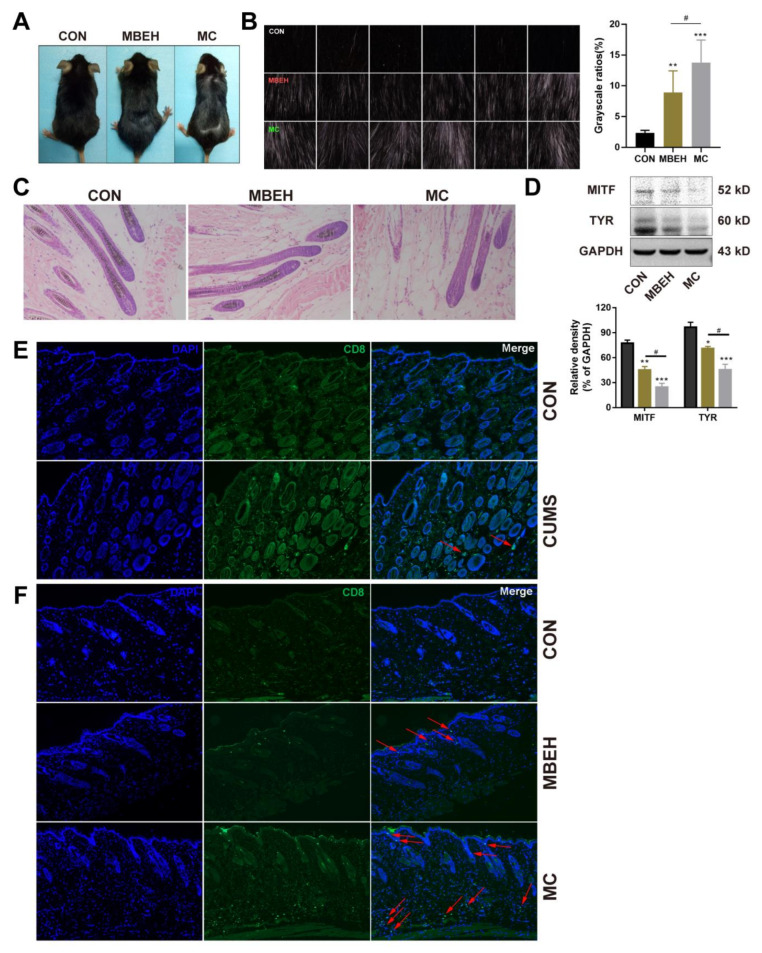
MBEH and MC inhibit skin melanogenesis. (**A**) Effects of MBEH and MC on melanin synthesis in the back skin of mice. (**B**) Effects of MBEH and MC on depigmentation of mouse dorsum. The phenotype of deep-layer hair pigment taken by the MPA 580 system in the modeling area on the mouse dorsum. Magnification: 10×. The grayscale was scanned by ImageJ to document the appearance of CON, MBEH and MC. (**C**) H&E staining of the mouse skin. Magnification: 200×. (**D**) The protein level of MITF and TYR were determined by Western blot analysis (n = 6). (**E**,**F**) Effects of CUMS, MBEH and MC on CD8^+^ T cell infiltration in skin measured by immunofluorescence assay. CD8 protein was labeled by an anti-CD8 alpha antibody conjugated IgG H&L (green). Cell nucleus was counterstained by DAPI (blue). Colocalization showed by red arrows were CD8^+^ T cells. Magnification: 100×. Results were shown as mean ± SD and were representative of three independent experiments. * *p* < 0.05, ** *p* < 0.01, *** *p* < 0.001, compared with CON; # *p* < 0.05, MC was compared with MBEH. Multiple groups were tested using one-way ANOVA.

**Figure 4 ijms-24-06990-f004:**
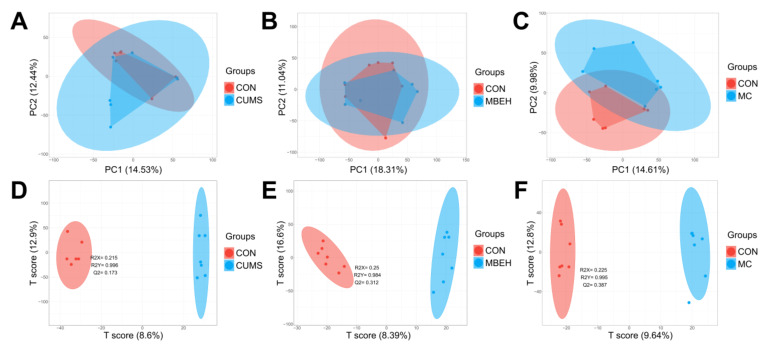
The PCA and sPLSDA score plots of each model. (**A**–**C**) The PCA score plot of CUMS, MBEH and MC mice. (**D**–**F**) The sPLSDA score plot of CUMS, MBEH and MC mice. Red represents the control group; blue represents the model group.

**Figure 5 ijms-24-06990-f005:**
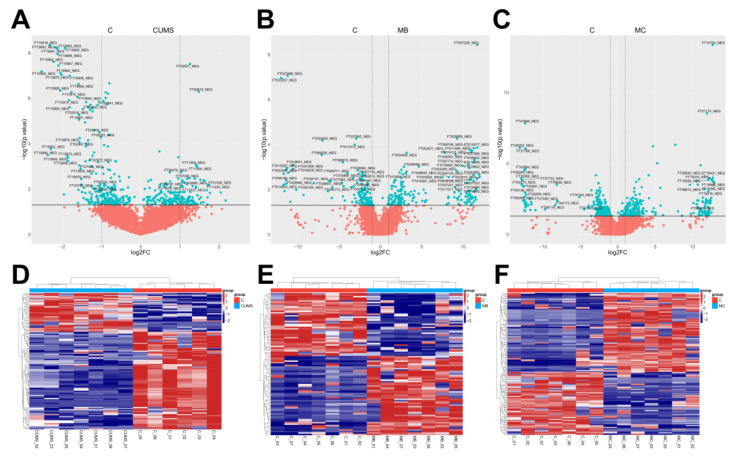
Volcano map scores and differential ion clustering heatmap analysis of each model. (**A**–**C**) Volcano map scores on the overall distinct metabolic signatures of the skin tissue of CUMS, MBEH and MC mice. (**D**–**F**) Differential ion clustering heat map analysis results of the overall metabolic profile of the skin of CUMS, MBEH and MC mice. Red represents the control group; blue represents the model group.

**Figure 6 ijms-24-06990-f006:**
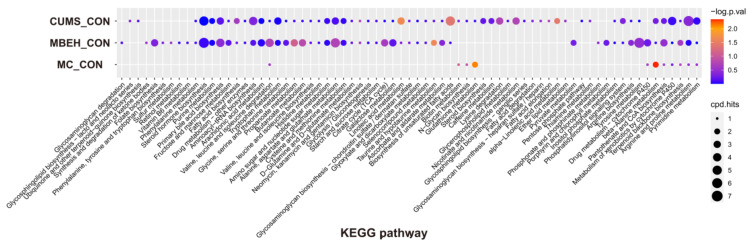
Analysis of KEGG enrichment of differential metabolites in the skin of CUMS, MBEH and MC mice.

## Data Availability

All data used to support the findings of this study are included within the article.

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
