# Peer review of "Optimization of Monobenzone-Induced Vitiligo Mouse Model by the Addition of Chronic Stress"

_ijms, 2023, doi:10.3390/ijms24086990_

Round 1

Reviewer 1 Report

The manuscript entitled ” Optimization of monobenzone-induced vitiligo mouse model by the addition of chronic stress” by Jing Dong et al focuses on the valuation of impact of chronic  stress in development mouse model  of vitiligo. The authors of the study, mental inducement added to the monobenzone (MBEH) induced vitiligo model and determined that chronic unpredictable mild stress (CUMS) inhibited the melanogenesis of skin.

The results of authors reflect the role of chronic mental stress in promoting depigmentation, and the establishment of a multi-factor induced animal model combined with chronic mental stress can has significance for the evaluation of vitiligo drugs.

The results obtained are significant for a better understanding of the mechanism of pathology in question.

The validity of the results obtained and method suggested is unquestionable.

The following improvements in the article would help other researchers to understand the significance of the results obtained in this work.

1.            The authors of the article claimed: As shown in Figure 1H, the expression levels of MITF and TYR proteins were significantly down-regulated after CUMS compared with CON. However, the Figure 1H their claim is not confirmed. The quality of presented western blot analyses are not satisfaction.

2.            The authors described interesting and effective model of developing pathology with CUMS. Nevertheless, the authors did not refer to a result of developing other pathologies with analogical CUMS.

3.            Text of 4 figure is too small, unreadable.

Author Response

Response to Reviewer 1 Comments

Point 1: The authors of the article claimed: As shown in Figure 1H, the expression levels of MITF and TYR proteins were significantly down-regulated after CUMS compared with CON. However, the Figure 1H their claim is not confirmed. The quality of presented western blot analyses are not satisfaction.

Response 1: Thanks for the kind reminder. We have modified the western blot image in Figure 1H. The image is one of the three duplicates in western blot and the renamed original image has been uploaded.

Line number in updated manuscript: Line 115.

Point 2: The authors described interesting and effective model of developing pathology with CUMS. Nevertheless, the authors did not refer to a result of developing other pathologies with analogical CUMS.

Response 2: Thanks for the kind reminder. We have supplemented more information in the part of Discussion.

Line number in updated manuscript: Line 248-255.

Point 3: Text of 4 figure is too small, unreadable.

Response 3: Thanks for the kind reminder. We have adjusted the text in Figure 4 and improved the clarity of the image.

Line number in updated manuscript: Line 235.

Reviewer 2 Report

The authors adequately performed the experiments, and the manuscript have written with enough data. So, I think this manuscript should be published in IJMS. However, I found a little downside in the manuscript. 

This manuscript is acceptable after adding some information.

according to the data of figure 5, 6

The authors performed sPLSDA and KEGG analysis. 

The methods of these assays are lack. Should be added in materials and methods.

Also, the data and the discussions are not enough. Authors should describe the data interpretation point by point. 

Author Response

Response to Reviewer 2 Comments

Point 1: According to the data of figure 5, 6, the authors performed sPLSDA and KEGG analysis. The methods of these assays are lack. Should be added in materials and methods.

Response 1: Thanks for the kind reminder. We have modified this part in the Materials and Methods.

Line number in updated manuscript: Line 384-414.

Point 2: The data and the discussions are not enough. Authors should describe the data interpretation point by point.

Response 2: Thanks for the kind reminder. We have supplemented more information in this updated manuscript.

Line number in updated manuscript: Line 267-269, 279-282, 285-297, 300-304.

Round 2

Reviewer 2 Report

Authors modified manuscript appropriately.

This manuscript is acceptable in present form.